# Clustering of the ζ-Chain Can Initiate T Cell Receptor Signaling

**DOI:** 10.3390/ijms21103498

**Published:** 2020-05-15

**Authors:** Yuanqing Ma, Yean J. Lim, Aleš Benda, Jieqiong Lou, Jesse Goyette, Katharina Gaus

**Affiliations:** 1EMBL Australia Node in Single Molecule Science, School of Medical Sciences, The University of New South Wales, 2052 Sydney, Australia; jieqiong.lou@unimelb.edu.au (J.L.); j.goyette@unsw.edu.au (J.G.); 2ARC Centre of Excellence in Advanced Molecular Imaging, The University of New South Wales, 2052 Sydney, Australia; 3ACRF Department of Cancer Biology and Therapeutics, The John Curtin School of Medical Research, The Australian National University, 131 Garran Road, 2601 Canberra, Australia; daniel.Lim@anu.edu.au; 4Research School of Electrical, Energy and Materials Engineering, College of Engineering and Computer Science, The Australian National University, 31 North Road, 2601 Canberra, Australia; 5IMCF at BIOCEV, Faculty of Science, Charles University, Průmyslová 595, 25250 Vestec, Czech Republic; ales.benda@natur.cuni.cz; 6School of Physics, University of Melbourne, Melbourne, VIC 3010, Australia; 7Department of Biochemistry and Molecular Biology, Bio21 Institute, University of Melbourne, Melbourne, VIC 3010, Australia

**Keywords:** TCR signaling, optogenetics, fluorescence correlation spectroscopy

## Abstract

T cell activation is initiated when ligand binding to the T cell receptor (TCR) triggers intracellular phosphorylation of the TCR-CD3 complex. However, it remains unknown how biophysical properties of TCR engagement result in biochemical phosphorylation events. Here, we constructed an optogenetic tool that induces spatial clustering of ζ-chain in a light controlled manner. We showed that spatial clustering of the ζ-chain intracellular tail alone was sufficient to initialize T cell triggering including phosphorylation of ζ-chain, Zap70, PLCγ, ERK and initiated Ca^2+^ flux. In reconstituted COS-7 cells, only Lck expression was required to initiate ζ-chain phosphorylation upon ζ-chain clustering, which leads to the recruitment of tandem SH2 domain of Zap70 from cell cytosol to the newly formed ζ-chain clusters at the plasma membrane. Taken together, our data demonstrated the biophysical relevance of receptor clustering in TCR signaling.

## 1. Introduction

T cell activation is initialized by peptide-bound major histocompatibility complex (pMHC) molecules engaging the αβ chain of the TCR. The TCR is constitutively associated with CD3 dimers —CD3εγ, CD3εδ, CD3ζζ—and upon ligation of the αβTCR, the immunoreceptor tyrosine-based activation motifs (ITAM) of CD3 molecules are phosphorylated by the tyrosine kinase Lck. TCR-CD3 phosphorylation is regarded as the first biochemically detectable signal of T cell activation and is termed TCR triggering. The exact mechanisms of how ligand binding to the extracellular domain of the αβTCR triggers CD3 phosphorylation of the intracellular motif remains unclear. Ligand binding may induce conformational changes [1,2] to facilitate coordinated re-arrangements of the TCR-CD3 subunits. However, conformational changes cannot explain TCR signaling induced by antibodies or triggering of simpler versions of the TCR such as chimeric antigen receptors (CARs). For example, T cells that express the extracellular and transmembrane domains of the co-receptor CD8 fused to the intracellular domain of ζ-chain respond to ligation in a similar manner as the TCR-CD3 complex, including activation of the canonical TCR signaling pathways [3]. The ability of CARs to elicit T cell responses analogous to the native TCR by integrating into the T cell signaling network has led to the clinical success of CAR T cell therapies [4]. It also raises the fundamental question of whether other mechanisms apart from conformational changes could facilitate TCR triggering.

Emerging evidence suggests that TCR clustering is required and/or regulates TCR triggering. First, while both monomeric and oligomeric pMHC bind to the TCR, only oligomeric ligands trigger TCR signaling, suggesting that ligand-induced receptor clustering is required for activation [5,6]. Other studies also suggested that antibody-induced receptor crosslinking was responsible for CAR triggering [3,7,8]. Second, it is well-documented that the TCR forms nano- and micro-clusters immediately after engagement with pMHC molecules or activating antibodies on cell or engineered surfaces [9,10,11,12]. In activated T cells, proximal signaling molecules such as Zap70, LAT, SLP-76 and PLCγ associate with TCR clusters [9,11]. A DNA hybridization-based CAR revealed that a single ligand receptor interaction can induce receptor clustering and initiate signaling [13]. Third, we found that only dense TCR clusters had a high signaling efficiency, suggesting a direct relationship between receptor clustering to signaling [14]. 

Although it is clear TCR clustering is important for early T cell activation, there is lack of direct evidence to show that receptor clustering alone is sufficient to induce TCR triggering. To test this, it is advantageous to bypass clustering by external ligand and directly induce receptor clustering. A previous study [15] employed a chemically induced oligomerization system to demonstrate that the induced clustering of the membrane-anchored ζ-chain was sufficient to trigger the activation of NF-AT, Oct/OAP and AP-1 transcription factors. In their system, the ζ-chain oligomerization was initialized by the membrane-permeable molecule FK1012 that induces homo-dimerization of FK506-binding protein (FKBP) in the cell. By comparing the activation of ζ-chain linked to one, two or three copies of FKBP, the authors concluded that valency of the receptor binding to FK1012 and the consequent oligomerization state was synergistic to receptor activation. 

To control clustering with higher flexibility, we developed an optogenetic ζ-chain construct that self-associates non-invasively within a few seconds upon irradiation with blue light and without the use of ligands. While optogenetic techniques are well-established in neuroscience, these tools have only just started to be applied to T cell biology. Two recent independent studies have employed optogenetics to control the ligand-receptor binding lifetime to test the kinetic proof reading model of T cell activation [16,17]. In the current study, we employed optogenetics to control the clustering of ζ-chain by tagging the photoreceptor cryptochrome 2 (Cry2) to the C terminus of ζ-chain. This construct forms homo-oligomers when the Cry2 chromophore is excited by blue light [18]. Using this tool, we demonstrated that the spatial clustering of the intracellular domain of the ζ-chain was sufficient to trigger downstream T cell activation signaling events. When light-induced ζ-chain clustering was induced in COS-7 cells, the presence of Lck was sufficient to initialize ζ-chain phosphorylation. While the results of the current study do not suggest that receptor clustering is the only or predominate mechanism of TCR triggering, we demonstrated here the biophysical consequence of TCR clustering and propose that ζ-chain clustering as a possible mechanism of TCR triggering. 

## 2. Results

### 2.1. An Optogenetic Tool to Control the Clustering of ζ-Chain 

We aimed to address whether clustering of the intracellular domain of ζ-chain is a regulatory element in TCR triggering. To avoid any association of the ζ-chain with endogenous TCR subunits, we replaced the extracellular and transmembrane domain of the ζ-chain with the membrane anchor of Lck so that the intracellular domain of ζ-chain is attached to the plasma membrane via palmitoylation and myristoylation groups [19]. Previous studies suggested that proteins with this membrane anchor diffuse freely in the plasma membrane [20,21]. To control the clustering of the ζ-chain non-invasively and without ligand binding, we linked the light sensitive photolyase homology region (PHR) domain (1-498 AA) of Cry2 [18] to the C-terminus of the ζ-chain (Figure 1a). Under blue light illumination, Cry2 self-associates into clusters within seconds in a reversible manner [22]. We fused the construct to mCherry to visualize the protein under 594 nm excitation and named the final construct ‘light-induced clustering of ζ-chain’ (LIC-Z, Figure 1a). As a control, we made the equivalent construct but without the Cry2 sequence (Figure 1b, named LIC-Z-delCry2). Both constructs expressed well in Jurkat or COS-7 cells and were mainly targeted to the plasma membrane. Confocal imaging confirmed that LIC-Z clustered immediately after irradiation with blue light while LIC-Z delCry2 did not (Figure 1c, Appendix A).

### 2.2. Clustered LIC-Z Induces Ca^2+^ Flux Independent of the TCR Complex

To verify if LIC-Z was signaling competent, we first investigated whether ζ-chain clustering was sufficient to trigger downstream signaling events, measured here as Ca^2+^ fluxes. We transfected LIC-Z into αβTCR-deficient T cells, Jurkat 76 cells, that have essentially no endogenous CD3 expression on the cell surface [23,24]. Thus, any signaling exhibited in these cells would be restricted to LIC-Z and would not involve other components of the TCR complex. A genetically encoded Ca^2+^ sensor, G-GECO [25], was co-transfected as a readout of T cell activation. Here, the 488 nm laser both excited G-GECO and activated Cry2, such that clustering of LIC-Z and time-lapse imaging of G-GECO was performed simultaneously. To confirm that the signaling was initialized by ζ-chain clustering, two control constructs were tested under identical conditions (Figure 2a): LIC-Z-delCry2, which lacks the Cry2 domain (Figure 1b) and is light insensitive and LIC-Z-Y-L, which has all six tyrosine residues in the three ITAMs of the ζ-chain replaced by leucine residues rendering it effectively a ζ-chain signaling-defective mutant. Time-lapse images (Figure 2b) and movies (Appendix A) showed that the clustering of LIC-Z caused Ca^2+^ influx in transfected Jurkat 76 cells ~80 s into irradiation with blue light (Figure 2c). In contrast, Jurkat cells expressing LIC-Z-delCry2 or LIC-Z-Y-L exhibited no measurable Ca^2+^ fluxes, suggesting that the observed Ca^2+^ signaling was triggered by ζ-chain clustering and required phosphorylated ITAMs. 

The canonical signaling pathway of TCR triggering follows a sequence of events that begins with the phosphorylation of ITAMs, followed by membrane recruitment of Zap70 to the phosphorylated ITAMs, where Zap70 becomes activated by both transphosphorylation [26] and phosphorylation by Lck, and the recruitment and tyrosine phosphorylation of LAT. We therefore enquired whether LIC-Z clustering engages the same signaling pathway. For this we repeated the Ca^2+^ flux experiment in Jurkat-derived cell lines lacking one of the proximal signaling molecules: JCam1.6 (Lck-deficient), P116 (Zap70-deficient), and a CRISPR/CAS9-gene edited LAT-knock out cell line. LIC-Z clustering did not induce Ca^2+^ flux in any of these cell lines (Figure 2d), suggesting that LIC-Z clustering is likely to trigger the canonical TCR activation pathway. To confirm this, we performed Western blotting on LIC-Z-transfected Jurkat 76 cell lines to examine the phosphorylation of typical downstream signaling molecules. Cells were irradiated for 45 s and kept in the dark for 1–5 min to prevent continuous LIC-Z clustering prior to cell lysis. We found that ζ-chain (at Y142), Zap70 (at Y319) and phospholipase C-γ1 (PLCγ, at Y783) were phosphorylated within the first minute after light exposure, and the extracellular signal regulated kinase (ERK1/2) after ~5 min (Figure 3). Activated PLCγ hydrolyses PIP2 to diacylglycerol (DAG) and inositol 1,4,5 trisphosphate (IP3), which releases Ca^2+^ from the endoplasmic reticulum and induces further flux through membrane Ca^2+^ channels [27]. It is thus likely that the observed Ca^2+^ flux was caused by PLCγ activation. ERK1/2 phosphorylation is required for the activation of T cell effector function such as interleukin-2 (IL-2) secretion [28]. Taken together, the data suggest that clustering of the cytosolic tails of ζ-chain at the plasma membrane of T cells is sufficient to initiate early TCR signaling in a similar manner as pMHC-TCR ligation. The need for the ITAM domains in LIC-Z and the lack of Ca^2+^ fluxes in Lck-deficient cells suggest that Lck is responsible for signal initiation. It is therefore possible that Lck phosphorylation of clustered substrates is more efficient than phosphorylation of non-clustered substrates. 

### 2.3. Lck Phosphorylated Clustered LIC-Z Efficiently in Reconstituted COS-7 Cells

To identify the minimal requirements for ζ-chain triggering and understand how clustering regulates ζ-chain phosphorylation, we reconstituted T cell signaling in a non-hematopoietic cell system, as done previously in other TCR signaling studies [29,30]. We co- transfected LIC-Z and wild-type Lck fused to green fluorescent protein (Lck GFP) into COS-7 cells and analyzed the LIC-Z ITAMs phosphorylation by immunostaining (Figure 4a). Since the constructs were contained on two separate plasmids, transfected cells contained polyclonal populations with a range of expressions of each construct (Figure 4a). For quantification, we therefore normalized phosphorylation levels to the expression levels of the LIC-Z construct (Figure 4b). In cells that expressed LIC-Z but lacked Lck expression, LIC-Z was not phosphorylated (Figure 4a red * labelled). In contrast, cells that expressed both constructs were stained by an Alexa647-tagged antibody against pY142 on ITAM3 (white cells in the merged image in Figure 4a). However, we found no difference in ITAM3 phosphorylation levels before and after light-induced clustering of LIC-Z (Figure 4b column 1 vs. 2), suggesting that in the absence of key T cell phosphatases such as CD45 and SHP-1, Lck activity was sufficiently high to phosphorylate ITAMs regardless of the clustering state of ζ-chain. However, we found that light-treated LIC-Z resulted in significantly more phosphorylation than light-treated LIC-Z-delCry2 (Figure 4b column 1 vs. 3), ruling out that light treatment per se impacted on ITAM phosphorylation. 

To reduce basal phosphorylation, we incubated COS-7 cells overnight with 25 µM of the Lck inhibitor PP2, which was washed out 1 min prior to light treatment. Our results showed that ITAM phosphorylation was substantially reduced compared to untreated cells (Figure 4b column 1 vs. 4). When LIC-Z clustering was induced with light in PP2-treated cells, the level of ITAM phosphorylation was increased significantly from 0.159 ± 0.012 to 0.244 ± 0.017 (column 4 and 5). In contrast, the non-clustering mutant LIC-Z-delCry2 displayed no significant change in phosphorylation under the same conditions (column 6 and 7). These experiments showed that firstly, when only LIC-Z was expressed in reconstituted COS-7 cells, there was neglectable amounts of phosphorylation by endogenous kinases. Secondly, when the kinase Lck was introduced, the basal kinase activity of Lck was sufficient to cause substantial LIC-Z phosphorylation. Thirdly, when Lck activity was controlled by PP2, clustering of LIC-Z facilitated increased phosphorylation by Lck. This suggests that under conditions where Lck activity is kept in check (either pharmacologically (Figure 4b) or via the presence of endogenous phosphatase in T cells (Figure 2), substrate clustering can be sufficient to shift the balance to a net increase in ITAM phosphorylation. 

### 2.4. LIC-Z Clustering Causes Cytosol to Plasma Translocation of Zap70 in COS-7 Cells

To measure ζ-chain triggering more directly, we used the tandem SH2 domain of Zap70 fused to mCherry (Zap70 tSH2) as a sensor for ζ-chain ITAM phosphorylation [31,32]. In the absence of ITAM phosphorylation, Zap70 tSH2 diffuses freely in cell cytosol. Upon receptor trigging, the sensor binds to phosphorylated ITAMs and thus translocates to the plasma membrane (Figure 5a). In agreement with the phosphorylation experiment, we noticed that in cells expressing higher levels of Lck, Zap70 tSH2 was already recruited to the cell plasma membrane prior to light activation of LIC-Z. Therefore, we conducted 3-color confocal imaging of Zap70 tSH2, Lck GFP and LIC-Z fused to YFP in COS-7 cells with medium or low Lck expression level (Figure 5b). By imaging the cross-section of cells, we could visualize and quantify the translocation of Zap70 tSH2 from the cytoplasm (center of cells) to the plasma membrane (edge of cells, Figure 5b and Figure 6a and Appendix A). To quantify Zap70 tSH2 association with the plasma membrane, we plotted the intensity ratio of regions representing the plasma membrane and the cytoplasm (Figure 5c). While the association of the LIC-Z-YFP to the plasma membrane was constant (Figure 5d), Zap70 tSH2 shifted from the cytoplasm to the plasma membrane upon ζ-chain clustering. The rapid association of Zap70 tSH2 with the plasma membrane after the start of imaging indicated that light-induced clustering enhanced ITAM phosphorylation above baseline levels.

To examine the role of Lck in Zap70 tSH2 recruitment to the membrane, we performed similar experiments using different Lck mutants and quantified Zap70 tSH2 membrane recruitment upon light-induced clustering of LIC-Z-YFP (Figure 5e). The SH2 domain in Lck facilitates interactions with phospho-tyrosine residues, including phosphorylated ITAMs [33,34]. To determine the contribution of these interactions, we used a Lck version with a mutation in the SH2 domain (R154K Lck). We reasoned that Lck transiently interacting with ζ-chain via the SH2 domain may enhance phosphorylation efficiency in clusters as it increases local Lck concentration and residence time. Indeed, expressing R154K Lck resulted in less Zap70 tSH2 recruitment after ζ-chain clustering compared to cells expressing wild-type Lck (Figure 5e and Figure A1a,b). This suggests that the SH2 domain of Lck was not mandatory but can enhance ζ-chain phosphorylation of clustered LIC-Z-YFP. Expressing an open but kinase-dead version of Lck, Lck K273R Y505F resulted in no increase in Zap70 tSH2 recruitment, indicating no change in ζ-chain phosphorylation levels (Figure 5e and Figure A1c,d).

After Zap70 tSH2 is recruited to the plasma membrane, it should bind to the phosphorylated LIC-Z and thus rearranged into clusters. By focusing directly on the plasma membrane adjacent to the coverslip, we could indeed observe the gradual formation of Zap70 tSH2 clusters in the membrane (Figure 6a and Appendix A). From the 3-color confocal images of the plasma membrane, we calculated the Pearson coefficient [35] for Zap70 tSH2 and LIC-Z-YFP co-localization (0.29 ± 0.03), which was substantially higher than the coefficient for Zap70 tSH2 and Lck co-localization (0.08 ± 0.02, Figure 6b). The latter result supports the notion that Lck-ζ-chain interactions are highly transient, as previously suggested [14]. To exclude the possibility that the double SH2 domain of Zap70 tSH2 outcompetes the single SH2 domain of Lck to bind to phosphorylated ITAM, we also performed Pearson coefficient analysis in cells expressing LIC-Z and Lck GFP without Zap70 tSH2, which displayed a lack of colocalization (Figure 6b lane 3). We followed the formation of LIC-Z-YFP clusters with live cell imaging and noticed that LIC-Z-YFP clusters were formed first, followed by the recruitment of Zap70 SH2 to LIC-Z-YFP clusters (Figure 6c), resulting in an increases in the Pearson coefficient over time (Figure 6d). 

In summary, the reconstitutions experiments demonstrated that the phosphorylation of ζ-chain ITAMs only required functional Lck and that ζ-chain phosphorylation was increased upon LIC-Z clustering, potentially because Lck could transiently interact with ζ-chain via its SH2 domains. 

### 2.5. Diffusion Analysis of Lck 

Because we could not detect clear Lck colocalization with LIC-Z clusters, we used diffusion measurements to map Lck interaction with non-clustered and clustered ζ-chain and reasoned that any intermolecular interaction would result in slower diffusion of Lck. To avoid emission crosstalk from GFP to mCherry channel as shown in Figure A2a, we performed single-point fluorescence spectral correlation spectroscopy as described previously [36]. The results showed that the diffusion times of Lck GFP and clustered LIC-Z were on millisecond and sub-seconds time scales for diffraction limited confocal spot. Cross-correlation analysis did not show any significant Lck to LIC-Z interaction (Figure A3a). However, due to the low signal-to-noise ratio in the data and cell heterogeneity, this finding does not exclude that a small population (i.e., less than 10%) do interact. 

Given the spatial heterogeneity within the plasma membrane, it was difficult to accurately focus the laser beam exactly at the LIC-Z clusters with single-point FCS. Thus, we used line-scanning fluorescence spectral correlation spectroscopy (line-scanning FSCS) [36,37] that allowed us to sample a larger section of the plasma membrane that containing both clustered and non-clustered regions. The overall line averaged spatial-temporal correlation data show that the diffusion of Lck cannot be fitted by a single component free 2D diffusion model (Figure A3b,c) as previously reported [38]. We therefore extracted the average Lck transition time crossing the illumination volume with a fitting free approach (Figure 7a and Figure A3d). Both single point FSCS (Figure A3a magenta line) and the kymograph plot (Figure 1f) have suggested that the LIC-Z clusters were stable during acquisition and largely immobile. This allowed us to compare the diffusion of Lck outside and inside the clusters. Each selected line scan was split into 20 segments and for each segment, we obtained the average diffusion time of Lck-GFP and intensity of LIC-Z, which was indicative of the location of LIC-Z clusters. The analysis showed that the diffusion coefficient of Lck was highly heterogeneous along the scanned line (black line of Figure 7b). The scatter plot of the intensity distribution of LIC-Z and Lck diffusion time showed no apparent correlation (Figure 7c), suggesting that Lck diffusion was not impacted by the presence of LIC-Z clusters. Similar results were obtained for signaling-incompetent LIC-Z-Y-L and monomeric LIC-Z-delCry2. This result implied that Lck did not form stable interactions with LIC-Z clusters, indicating that phosphorylation most likely arose from transient interactions. 

## 3. Discussion

We designed an optogenetic tool to test how ζ-chain clustering per se impacts on TCR signaling. LIC-Z did not trigger signaling via endogenous components of the TCR-CD3 complex since the transmembrane domain of ζ-chain that is required for interaction with other subunit [39] was replaced by the membrane anchor of Lck (Lck10). Several lines of evidence suggest that LIC-Z did not spontaneously self-associate into homodimers, higher order oligomers or clusters. First, the formation of CD3 ζ-ζ dimer is known to be coordinated through hydrogen bonding of residues on its transmembrane domains [40]. Second, there is evidence that the Lck membrane anchor does not support clustering. A previous study showed that co-expression of Lck10-GFP and Lck10-mCherry that attached to the plasma membrane by the same membrane anchor did not interact as indicated by a lack of Förster resonance energy transfer (FRET) [21]. Further, single molecule tracking studies have shown that the diffusion of Lck10-anchored proteins display free diffusion [20,41]. We therefore used LIC-Z to compare non-clustered to clustered states of the ζ-chain. Light-induced clustering of LIC-Z in Jurkat cells showed that spatial clustering of the intracellular domain of ζ-chain was sufficient to trigger signaling events. This was particularly obvious when comparing the effect of light-treated LIC-Z to light-treated LIC-Z-delCry2 that did not cluster. In reconstituted non-hematopoietic cells, we showed that Lck can phosphorylate non-clustered ζ-chain. When basal Lck activity was kept in check, the level of phosphorylation was further increased upon ζ-chain clustering. In T cells, where the basal level of phosphorylation is naturally suppressed by phosphatase activity, clustering of ζ-chain was sufficient to shift the kinase-phosphatase balance and induce signaling that resembles TCR triggering. 

The results of the current study suggest that ζ-chain could be a biophysical mechanism of TCR triggering. Importantly, this mechanism does not contradict or exclude other TCR triggering mechanism such as ligand-induced conformational change model [1,2,42,43] or the ‘safety-on’ model [44] of T cell activation. Both models predict that an additional extension of the intracellular CD3 motifs from plasma membrane upon ligand binding. The conformation model proposes that the force of ligand receptor binding is transmitted as a torque from the extracellular to the intracellular signaling motifs [45] facilitated by the tight and rigid association of the TCR αβ-chains to the CD3 signaling chains. The conformational change releases binding site for adaptor protein Nck [1] that is required for downstream signaling. The safety-on model suggest that the conformational change is initialized due to local increase of ionic strength that screens the charge at the membrane inner leaflet [46,47]. Molecular simulation has suggested that the ζ-chain is associated to the plasma membrane in a dynamically equilibrium manner [48] rather than static on/ off states. It is highly possible that such dynamic equilibrium is altered upon clustering, where the states become more stable through lateral crowding effects. It should be noted, however, that previous studies suggest that ITAMs could be phosphorylated by Lck in membrane-bound state [49], and fully phosphorylated ζ-chain could remain membrane-associated due to the clusters of positively charge motifs on the ζ-chains [50]. Whether clustering-mediated ITAM phosphorylation requires conformational changes and/or membrane detachments remains to be investigated.

Clustering-induced TCR triggering observed with LIC-Z is likely shares similar triggering mechanism as previous studies that use oligomeric forms of the soluble ligands to induce TCR activation [5,6,51]. It is also well documented that antibody cross-linking of single chain chimeric receptor of the ζ- or ε-chains of TCR, or the γ-chain of Fc receptor alone was sufficient to trigger downstream activation responses [3,52,53]. Increased ITAM phosphorylation upon ζ-chain clustering has also been reported in in vitro systems that containing ζ-chain, Lck and CD45 [54,55]. However, in these experiments, ζ-chain phosphorylation may have been driven by the transphosphorylation of Lck or exclusion of CD45 from clustered ζ-chains. Here, we provided additional insight that clustered ζ-chain is a more efficient substrate for Lck than non-clustered ζ-chain if basal Lck activity is kept low. Thus, the enhanced efficiency of Lck and exclusion of CD45 in TCR clusters could work synergistically in T cells, where basal ITAM phosphorylation levels and Lck activity are controlled by phosphatases such as CD45 [56] and the kinase Csk [57], respectively. However, given the reductionist systems used here, we would like to emphasize that ζ-chain clustering may only represent one of many possible TCR triggering mechanisms that could co-exist in T cells. Co-existing TCR triggering mechanisms may allow T cells to sense and respond to antigens under vastly different conditions. For example, it has been reported that TCR can be activated by monomeric pMHC or monomeric TCR binding Fab antibody when the ligand is anchored to lipid surface [58,59,60,61], and the downstream signaling profile differs from antibody crosslinking in solution [59]. Emerging evidence suggesting that monomeric ligand receptor interaction can induce receptor clustering [13], and it is the newly formed rather the preformed higher ordered receptor configuration drives the signaling [62]. 

Currently, it is not known why Lck phosphorylation of ITAM domains occurs more efficiently in ζ-chain clusters. We found no evidence of stable interactions between Lck and clustered LIC-Z. The time Lck spent in the clustered regions was comparable to that spent in non-clustered regions of LIC-Z, suggesting that Lck diffusion was not slowed down by LIC-Z clusters. It remains possible that Lck interacts more frequently or for longer with clustered LIC-Z via its SH2 domain but in our experiments, these changes were not detectable. A recent study showed that Lck interact selectively with ε-chain through charge to charge interactions [63]. Thus, it is possible that Lck can forms more stable interactions with TCR complex clusters through association to ε-chain although we previously did not find evidence for TCR-Lck co-clustering with single molecule localization microscopy [14]. 

In summary, we showed that the spatial clustering of the intracellular motif of the ζ-chain was sufficient to trigger proximal TCR signaling events that are reminiscent of canonical TCR activation. Lck-mediated phosphorylation of clustered ζ-chain may be a distinct TCR triggering mechanism that is independent of phosphatase activity but requires a low basal level of ITAM phosphorylation. This TCR triggering mechanism is not exclusive of other TCR triggering mechanisms and indeed, may co-exist with other process that induce and regulate TCR signal initiation and propagation.

## 4. Materials and Methods 

### 4.1. Plasmids

All constructs and mutants used in the current study were prepared by standard subcloning, overlapping PCR and site-directed mutagenesis. The primers used and the sequence of LIC-Z and related mutants are provided in Appendix B and deposited in Addgene (153543-153546). The Y to L mutations of the ζ-chain intracellular domain was subcloned from a ζ-chain-6YL-PSCFP2 construct [14]. The G-GECO (#32447) and Zap70 tSH2 (#27137) constructs were obtained from Addgene, Watertown, MA, USA. 

### 4.2. Cell Culture and Light Treatment

COS-7 (ATCC CRL-1651) cells were cultured in Dulbecco’s Modified Eagle’s medium (DMEM) with 10% fetal bovine serum (FBS). Jurkat 76 [14], JCam1.6 (ATCC CRL-2063), Jurkat P116 [64], Jurkat LAT KO (Gift from Jieqiong Lou) cell lines were all cultured in Roswell Park Memorial Institute medium (RPMI) supplemented with 10% FBS and L-Glutamine. Cells were cultured in incubators set at 37 °C with 5% CO_2_. All cell lines were tested and confirmed to be Mycoplasma negative. Transient cell transfection was conducted using an electroporator (Invitrogen Neon, Sydney, Australia) following the manufacturer’s protocols and all cells were imaged or treated 18–24 h post transfection. For light activation of LIC-Z transfected cells, the cells were washed, and light activated in HBSS buffer containing Ca^2+^ and Mg^2+^ (Thermo Fisher, #14025076, Sydney, Australia). For activation on the microscope, the activation was done using a 458 or 488 nm laser. For activation of a large cell population for Western blot, light activation was done using a 29 mW/cm2 470 nm LED blue light illuminator box (Maestrogen, #LB16, Taipei, Taiwan) (measured at 488 nm using a Newport photometer). 

To measure background phosphorylation levels in COS-7 cells, 25 µM of PP2 (Sigma Aldrich, #p0042, Sydney, Australia) was added to the cell culture media following cell transfection and incubated overnight. The cells were switched to PP2-free HBSS buffer for 1 min prior to light treatment to allow for the recovery of Lck kinase activity.

### 4.3. Western Blotting

Transfected and wild type Jurkat 76 cells were pelleted by centrifugation at 1500 rpm for 5 min and resuspended in 150 µL of HBSS buffer containing Ca^2+^ and Mg2+. The samples were either kept in the dark or light-activated using the LED blue light illuminator box for 45 s, followed by further incubation (0–5 min) in the dark. All cell treatments here were performed at room temperature (23 °C). After incubation, cells were quickly lysed in 8% Lithium dodecyl sulfate LDS loading buffer preheated at 85 °C (Invitrogen Cat#NP0007 Sydney, Australia, and incubated with stirring at 500 rpm for 5 min at 85 °C (Eppendorf, ThermoMixer C, Sydney, Australia). Lysates were either sonicated (Sonifier 250, Branson, Emerson, Ferguson, MO, USA) or homogenized with a Bio-Gen PRO200 homogenizer with an attached 5 × 75 mm flat bottom generator probe (Pro-Scientific, Monroe, CT, USA) and then loaded into 4–12% Bolt Bis-Tris gels (Invitrogen) for gel electrophoresis. Resolved proteins were transferred onto a PVDF membrane (IB24001) using the iBlot2 dry transfer system (Life Technologies, Sydney, Australia).

For Western blotting, membranes were blocked with 5% (*w*/*v*) of BSA for 1 h, followed by overnight incubation in primary antibodies against pERK (1:2000, Cell Signaling, Cat# 9101S, Danvers, MA, USA), pZAP70 (pY319, 1:1000, Cell Signaling, Cat# 2701S), pCD3ζ (pY142, 1:1000, abcam, Cat# ab68235, Danvers, MA, USA), pPLCγ (pY783, 1:1000, Cell Signaling, Cat#2821s), GAPDH (1:5000, Abcam, Cat# ab8245) at 4 °C in 4% BSA. Blots were then incubated in HRP-linked secondary antibody against mouse or rabbit IgG (1:3000, Cell Signaling, Cat# 7074S and 7076S) in 4% BSA for 1 h at room temperature. Protein bands were visualized by chemiluminescence (Pierce ECL Western Blotting Substrate, Thermo Scientific) following the manufacturer’s protocol and then imaged using ImageQuant LAS4000 Mini gel documenter (GE Life Sciences, Marlborough, MA, USA). Protein band intensity was analysed in ImageJ (National Institutes of Health, Bethesda, MD, USA) by taking the integral of the pixel intensity of the protein bands. 

### 4.4. Imaging

Cells were imaged in either a Leica SP5 or Zeiss 880 laser scanning Confocal microscope. For multicolor imaging, sequential scanning mode was used to avoid crosstalk during the collection of emitted fluorescence. For Ca^2+^ imaging of Jurkat cells, transfected cells were washed and imaged in suspension in HBSS buffer containing Ca^2+^ and Mg2+ at room temperature through an HCX APO L 20×/1.0 NA water immersion objective. G-GECO and LIC-Z were sequentially excited by 458 nm and 594 nm lasers, and fluorescence emission was collected between 470–540 nm and 590–670 nm spectral bands, respectively. For Zap70 COS-7 live cell experiments, transfected cells were washed and imaged in HBSS buffer containing Ca^2+^ and Mg^2+^ using the Zeiss 880 with an LD C-Apochromat 40×/1.1 NA water immersion objective at room temperature. GFP, YFP and mCherry that tagged to Lck, LIC-Z-YFP and Zap70 tSH2, respectively were excited by 458, 514 and 594 nm lasers, which were sequentially reflected to the objective through the triple dichroic mirror (458/514/594 nm). Two PMTs and one GaAsP detector were gated to 460–508, 526–580, and 605–721 nm to sequentially capture the corresponding fluorescence. For pCD3z immunostaining, COS-7 cells co-transfected with LIC-Z or LIC-Z-delCry2 with Lck GFP were light treated/not treated and fixed in 4% paraformaldehyde and permeabilized with 0.05% Triton X-100. The cells were blocked with 4% BSA and stained with a 1:200 (*v*/*v*) dilution of Alexa Fluor 647 Mouse anti CD247 pY142 antibody (BD Bioscience #558489, Franklin Lakes, NJ, USA) for 1 h at room temperature. The stained cells were washed and imaged on the Leica SP5 with sequential excitation by a 488, 561 and 633 nm pulsed white light laser directed at an HCX PL APO CS 20× 0.70 IMM objective through of 488/561 nm dichroic mirror and 633 nm notch filter. Two hybrid HyD and one PMT detectors were spectrally gated to 497–543, 569–630, and 652–774 nm to sequentially capture the corresponding fluorescence. A line average of 32 lines was used during image acquisition to increase the signal to noise ratio of the image. The two color imaging of LIC-Z and Lck GFP in COS-7 cells were acquired under Airyscan mode on Zeiss880 microscope. The tagged GFP and mCherry fluorophores are excited frame by frame sequentially using 458 and 594 nm lasers that reflected from the triple dichroic mirror (458/514/594 nm) to the Plan-Apochromat 63×/1.4 Oil DIC M27objective. The emission collected travel through the same dichroic mirror and a dual emission band pass filter (bandpass 495–550 nm/LP 570) prior arrive to the Airyscan fibre head, which were set to super-resolution mode under Zen software. The diameter of the hexagon was configured to be equivalent of 1.26 Airy unit. The super-resolution image was reconstructed using Zen software prior to colocalization analysis using self-written MATLAB code.

### 4.5. Image Analysis

The majority of the image analyses were conducted in custom wrote MATLAB code (Mathworks). The script is available online at (https://github.com/mayuanqing8/Light-induced-CD3-clustering-project). 

The standard deviation STD of pixel intensity inside the ROI as shown in Figure 1e is done in imageJ, where the STD value of the same ROI is calculated over each ROI and plotted as a function of time as shown. The kymograph is plotted from a line of ~6µm selected from the same time lapse images. 

The Ca^2+^ profile of individual cells shown in Figure 2c were extracted from the time-lapse images by manually selecting 20 regions of interest (ROI) and plotted as a function of time in ImageJ. The increase of Ca^2+^ shown in Figure 2d was quantified as the normalized increase of G-GECO intensity (average of the last 6 frames subtracted by the average of the first 6 frames divided by the average of the first 6 frames).

The quantification of pζ-AF647 immunostaining shown in Figure 4 was performed as follows: Since not all the cells were co-transfected with LIC-Z and Lck GFP, the first step was to identify the cells containing both constructs. This was done by logical AND selection of pixels in both LIC-Z and Lck GFP channels that had intensity values greater than the ostu threshold value of the respective channel. This produced a mask to select cells that contained both constructs. The same mask was used to extract the intensity values of the pζ-AF647 channel. In order to correct for the intensity variation of pζ-AF647 due to difference of protein expression level between LIC-Z and LIC-Z-delCry2, the extracted mean intensity value of pζ-AF647 was divided by the mean intensity value of LIC-Z or LIC-Z-delCry2. In order to correct for intensity variation due to human error, such as change of objective focus, or cell staining efficiency, the background intensity of pζ-AF647 from cells that contained only LIC-Z with no Lck GFP was used to standardize pζ-AF647 intensity across different sample groups. 

The cytosol to plasma membrane translocation of Zap70 tSH2 shown in Figure 5c is analyzed as follows: briefly, ROIs containing single cells were first manually cropped from the time-lapse movies using ImageJ and saved as hyperstack Tiff images, which are load into MATLAB. The borders of each cell were identified by Sobel edge detection, followed by a number of clean-up steps including image dilation, filling interior holes, and smoothing of the detected object. An ascending (0–1) distance to center 2D matrix was created from the center of mass to identified edge of cells. Masks for cell plasma membrane and cytosol was created by logical selection of matrix values between 0 to 0.25 and 0.25 to 1, respectively. The fluorescence intensity of the Zap70 tSH2 in the selected regions was extracted and analyzed. To verify the code works, the same mask was used to extract intensity change of permanently membrane attached LIC-Z, which has showed no change as shown in Figure 5c and Figure A1.

For image Pearson coefficient analysis shown in Figure 6b,d, the images were first thresholded using the otsu [35] method to remove background regions as Pearson coefficient analysis method is known to be sensitive to background noise [65]. The image Pearson coefficient was calculated as
(1)∑ (Gi−G¯)(Ri−R¯)∑ (Gi−G¯)2∗∑ (Gi−G¯)2
where *G_i_*, *R_i_* and $\bar{G}$ and $\bar{R}$ are incident and mean pixel intensity values of green and red channels of the background removed regions. 

### 4.6. Fluorescence Spectral Correlation Spectroscopy 

Both the single point FSCS and line-scanning FSCS was acquired using the Zeiss 880 laser scanning confocal microscope. Then, 488 and 594 nm CW lasers were focused on the Plan-Apochromat 63 × 1.4 oil objective through a dual color dichroic mirror (488/594 nm) to simultaneously excite GFP and mCherry at the basolateral membrane of the cell. The closer distance of detection volume from the coverslip helps to reduce optical distortion related to refractive index mismatch, affecting the correlation curve [66]. The fluorescence emitted was collected by the same objective and dichroic mirror prior arrive to the GaAsP detector set at either 6 spectral channels (495–510, 511–530, 531–560, 561–610, 611–630 and 631–680 nm) for single-point FSCS or 10 spectral channels (481–499, 500–518, 519–538, 539–557, 558–576, 577–595, 596–616, 617–635, 636–654 and 655–673 nm) in single photon counting detection mode. For line-scanning FSCS, the scanning distance was 5.355 micrometers over 256 pixels with a pixel size of 21 nm. The laser was scanned in a bidirectional manner over the same one-dimensional space at a speed of 0.51 μs per pixel, and 154 μs line scan time. The quantity of 300,000 lines were collected over a time period of 47 s. The data was saved and converted into TTSR file format in a custom made LabView software [36]. 

The analysis of the single point FSCS data was performed as previously published [36], and the line-scanning FSCS data was analyzed similar to the single point FSCS analysis with some modifications. Briefly, the pixel and spectral channel identity of each photon collected is used to retrieve where the photon was detected from. The assumption was that in any given time, the intensity of any spectral channel was attributed from a linear combination of the fluorescent species present in the excitation volume. Based from the emission spectra of each reference dye (the spectra referred is the dichroic mirror filtered spectra) and the dye mixed sample spectra, one can estimate the contribution from respective dye species by applying a spectral filter fgi and fri to the sample spectra, fgi and fri are the calculated filter for GFP (g stand for green) and mCherry (r stand for red), respectively. Essentially, the product of the sample spectra with the calculate filter should reproduce the reference dye spectra and its contribution to the mixed sample spectra. Here the spectra are the intensity normalized spectra, and the sum of the two filters is equal to 1. The filters are recalculated for each sample and treated as a constant value in the correlation analysis. The calculated filters are integrated into the autocorrelation algorithm to weight the photons depend on the spectral channel to recover the correlation that attributed from individual dyes. The cross-correlation is calculated as:(2)Ggr(τ)=〈∑ fgi∗Ij(t)∗∑ fri∗Ij(t+τ)〉〈∑ fgi∗Ij(t)〉∗〈∑ fri∗Ij(t)〉.

Here 〈 〉 notes averaging over all time *t*. f_gi_ and *f_ri_* are calculated filters applied to green and red channel respectively. *I_j_(t)* is the photon intensity of spectral channel *j* at time *t*. In case of autocorrelation, filter f_gi_ = f_ri_. Compared to traditional single-point FCS, the sampling rate of line-scanning FSCS is slower and defined by the pixel dwell time of 0.51 μs. Given the deadtime of the GaAsP detector is about 100 ns, the probability of multiple photon event during this time period is considerable. Under the single photon counting mode, single photon events are registered as a pixel intensity of 1, where multiple photon event re registered 2, 3… and so on, from 0 to 255 as 8-bit Tiff images. Higher weighting factors (scaling the filter by the intensity value) has to be applied to those multiphoton pixels so that their contribution to the correlation calculation are scaled accordingly. 

The fitting of autocorrelation curves was performed as previously described [37]. Here, the autocorrelation took a two-dimensional spatial and temporal correlation form, where the Gaussian shaped lateral spread in the scanning axis was calculated by the spatial lateral cross correlation of the neighboring pixels along the line at G(0). The Gaussian here is equivalent to the cross section of the product of the excitation and detection PSF of the microscope [37]. The FWHM of the Gaussian curve from the spatial cross-correlation was used to conveniently retrieve the diameter w of the detection volume, so that no calibration using a reference dye is needed. Orthogonal to the plane of the Gaussian form was the temporal autocorrelation curve that can be analyzed with model fitting or fitting free options. The intensity fluctuation of 256 pixels along the line can be split into 20 segments each of a size of 250 nm that equivalent of 20 raw single-point FCS measurements. For model fitting of the correlation curves, the autocorrelation curves of the 20 segments are averaged to increase statistical robustness. Here, the resulting 2D autocorrelation curve can be fitted to the two-dimensional n species free diffusion model as shown in Figure A3b,c.
(3)$$G\left(t,\delta \right)=G_{offset}+\sum^{\;}A_{i}*\frac{1}{4D_{i}t+w^{2}}e^{\left(\frac{-\delta^{2}}{4D_{i}t+w^{2}}\right)}$$
where *t* is the temporal correlation lag time, and δ is spatial correlation lag. *A_i_* and *D_i_* are the autocorrelation amplitude and diffusion coefficients of the respective species. *w* is the beam waist of excitation laser. 

For investigating the heterogeneity of Lck GFP diffusion along the scanning line, the 20 raw auto-correlation curves along the line were analyzed individually. Given that a single species diffusion model was insufficient to fit Lck GFP autocorrelation curves as shown in Figure A3b,c, and the signal to noise ratio of the segmented single-point autocorrelation curves are too low for accurate two species model fit. We used a fitting free method as shown in Figure A3d and Figure 7a, where data of each segment was subject to independent temporal correlation analysis. Cubic spline smoothing was applied to each correlation curve to extrapolate to zero lag time to estimate G(0). The G(0) value was used as a reference to interpolate the half lag time, who’s amplitude is 50% of the value of G(0). This is defined as the transition time γD of the molecule across the excitation volume. Here Lck GFP diffusion coefficient is calculated as D=w24∗γD. The beam waist w value was obtained from the spatial correlation analysis as mentioned above. In this fitting free analysis, there was no assumption made relating to the number of diffusion component or diffusion mode. 

### 4.7. Data Availability

The source data that support the findings of this study are openly available in Dryad at: https://datadryad.org/stash/share/ikWboR0EB0_7jbpCFntYpIrAJ4y7deGpva8JgHRiVjA; please note that 7-zip is needed to extract the file for windows system.

## Figures and Tables

**Figure 1 ijms-21-03498-f001:**
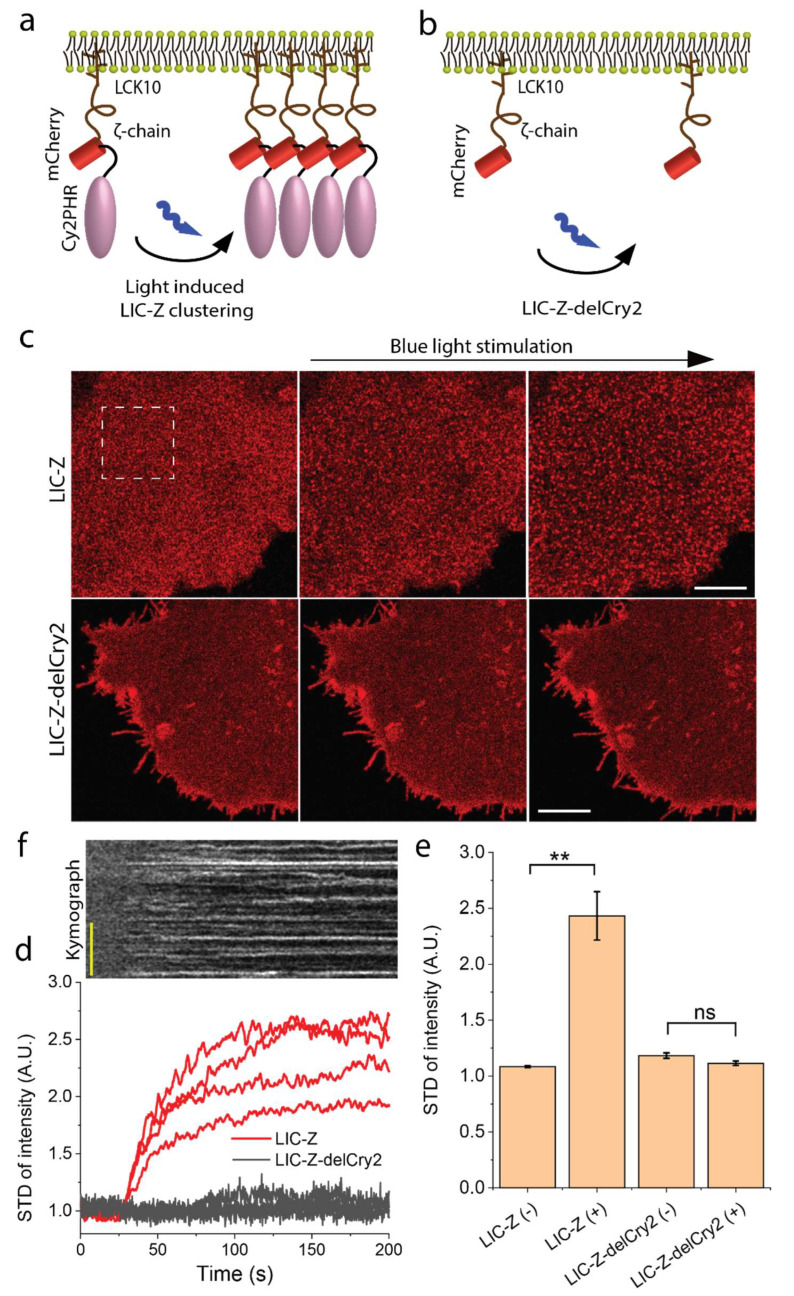
Light-induced clustering of ζ-chain. (**a**) Schematic of the light-induced construct, termed LIC-Z, consisting of a membrane anchor (first 10 amino acids of Lck), the cytosolic tail of ζ-chain, mCherry and the light sensitive domain PHR of Cry2. Prior to illumination, LIC-Z diffuses freely in the plasma membrane. Upon illumination with blue light, LIC-Z self-associates into clusters. (**b**) Schematic of the control construct, termed LIC-Z-delCry2, which lacks the Cry2 domain and is light insensitive. Schematics (**a**,**b**) are not to scale. (**c**) Confocal images of LIC-Z (top row) and in LIC-Z-delCry2 (bottom row) in COS-7 cells. Cells were irradiated with low intensity 458 nm laser light as indicated and continuously imaged by exciting mCherry with 594 nm laser light. Regions of interest (ROIs) in dashed box were selected for quantitative analysis of the clusters in d, e. The size of the dashed box is 8 µm. Scale bar = 6 µm. (**d**,**e**) Representative traces (**d**) and quantification (**e**) of the standard deviation of pixel intensity in selected ROIs as indicated by dashed box in (**c**) of light treated LIC-Z and LIC-Z-delCry2 prior and post 458 nm laser irradiation as indicated by (−) and (+) in the axis label. (**f**) Representative kymograph of light-induced LIC-Z clustering, with the identical time scale as in (**d**). The 458 nm light was switched on at 25 s. Images in (**c**) are representatives of n = 5 experiments. In (**e**), data are mean and standard error of *n* = 15. ** *p* < 0.001 (one-way ANOVA with Fisher LSD post hoc test). In (**f**), scale car = 2 µm.

**Figure 2 ijms-21-03498-f002:**
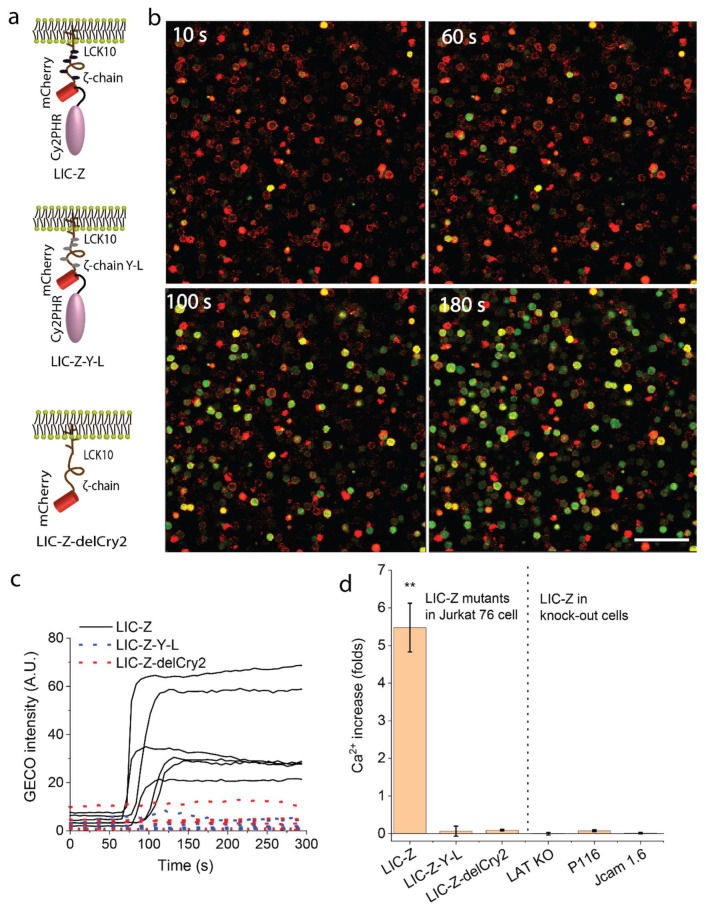
LIC-Z clustering induces Ca^2+^ flux in Jurkat cells. (**a**) Schematics of LIC-Z (top), signaling incompetent LIC-Z-Y-L (middle), and light insensitive LIC-Z-delCRY2 (bottom). (**b**) Confocal images of Ca^2+^ flux in Jurkat 76 cells co-transfected with LIC-Z (red) and Ca^2+^ sensor G-GECO (green). Images were taken at the indicated time points after irradiation with blue light. Scale bar = 150 µm (**c**) G-GECO intensity traces over time for single cells expressing LIC-Z (solid line), LIC-Z-delCRY2 (red dotted line) and LIC-Z-Y-L (blue dotted line). (**d**) Quantification of Ca^2+^ flux, as fold increase over baseline level, in Jurkat 76 cells expressing LIC-Z, LIC-Z-Y-L and LIC-Z-delCry2, and LIC-Z expressed in Jurkat cells deficient of LAT (LAT KO), Zap70 (P116) or Lck (JCam 1.6). In (**d**), data are mean and standard error of *n* = 30 cells. ** *p* < 0.001 between the first column to the rest of all columns (one-way ANOVA with Fisher LSD post hoc test).

**Figure 3 ijms-21-03498-f003:**
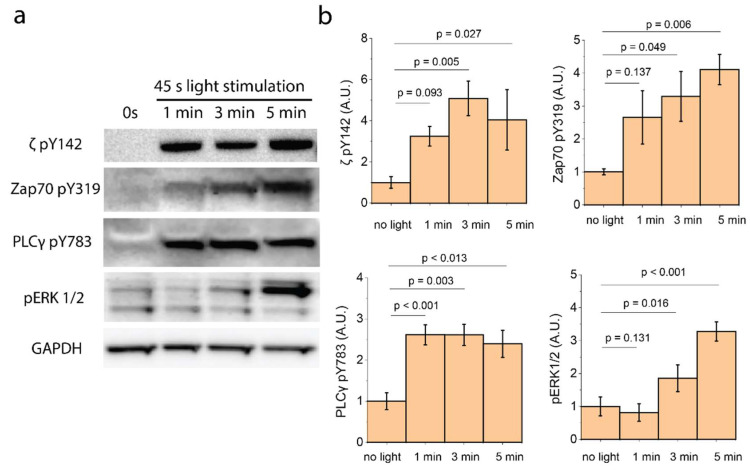
Western blot of T cell receptor (TCR) proximal signaling molecules in LIC-Z-transfected Jurkat 76 cell line. Western blot images (**a**) and quantification (**b**) of Jurkat 76 cell line transfected with LIC-Z and treated, or not, with blue light for 45 s followed by incubation in the dark for indicated time period (minutes). GAPDH was used as a loading control. Data are from ≥3 biological replicates. One-way ANOVA with Fisher LSD post hoc test.

**Figure 4 ijms-21-03498-f004:**
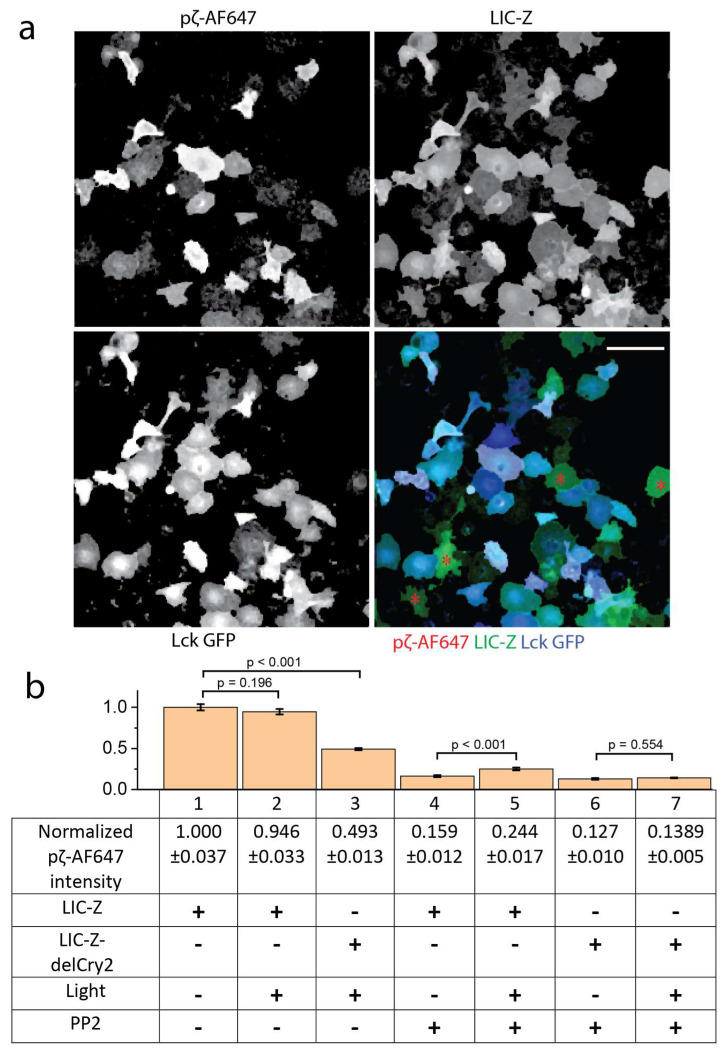
ITAM phosphorylation of LIC-Z and LIC-Z-delCry2 in COS-7 cells. (**a**) Representative intensity images of individual and merged channels of COS-7 cells co-transfected with LIC-Z (green) and Lck-GFP (blue) and immune-stained with Alexa647-tagged antibodies recognizing Y142 in ζ-chain (red). Red asterisks (*) labelled cells that expressed only LIC-Z and exhibited no pζ-AF647 staining. Scale bar = 150 µm. (**b**) Bar graph and table of pζ-AF647 intensity normalized to LIC-Z or LIC-Z-delCry2 expression levels (+/−) in cells that were treated or not treated with light (+/−) and were pre-treated or not treated with 25 µM of PP2 (+/−). Normalized pζ-AF647 intensity in cells expressing LIC-Z without light or PP2 treatment were set to 1. One-way ANOVA with Fisher LSD post hoc test.

**Figure 5 ijms-21-03498-f005:**
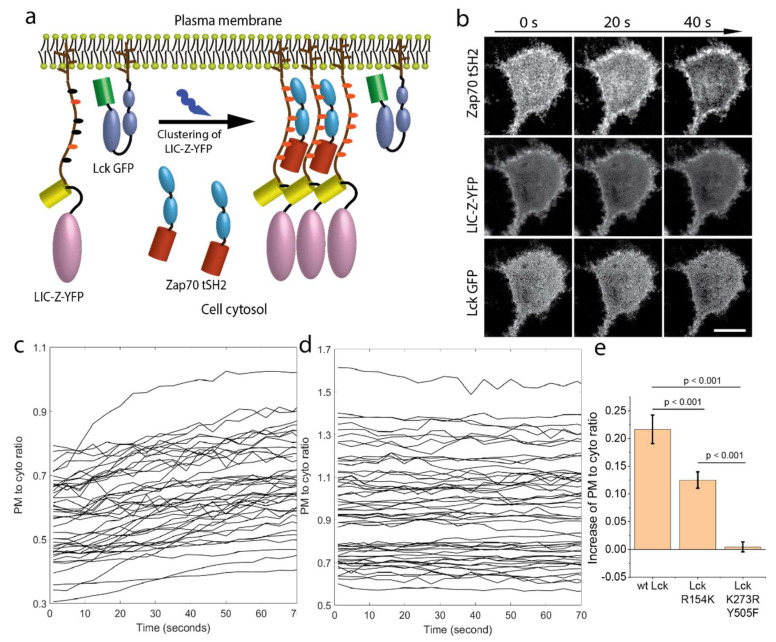
Reconstituting and imaging ζ-chain triggering in COS-7 cells. (**a**) Schematics of reconstituting ζ-chain triggering in COS-7 by co-expressing LIC-Z-YFP, Lck GFP and the mCherry-tagged tandem SH2 domain of Zap70 (Zap70 tSH2). Prior to ζ-chain triggering, Zap70 tSH2 diffuses freely in the cytosol. Upon light-induced clustering, ζ-chain triggering is read-out as the translocation of Zap70 tSH2 from the cytosol to plasma membrane since Zap70 tSH2 binds specifically to phosphorylated ITAMs on the LIC-Z-YFP. (**b**) Representative live cell confocal images of light-induced clustering of LIC-Z-YFP (top row), distribution of Zap70 tSH2 (middle) and Lck GFP (bottom). Scale bar = 20 µm. (**c**,**d**) Quantification of Zap70 tSH2 (**c**) and LIC-Z-YFP (**d**) intensity at the plasma membrane relative to the cytosol immediately after irradiation. Each line represents one cell. (**e**) Quantification of Zap70 tSH2 recruitment to the plasma membrane for cells expression wild-type Lck, Lck with a mutation in the SH2 domain (Lck R154R) or open and kinase dead Lck (Lck K273R Y505F) prior and post light induced clustering of LIC-Z-YFP. Zap70 tSH2 intensity ratio is shown as a fold-change relative to the first four imaging frames. Data are mean and standard error of *n* = 40 cells. One-way ANOVA with Fisher LSD post hoc test.

**Figure 6 ijms-21-03498-f006:**
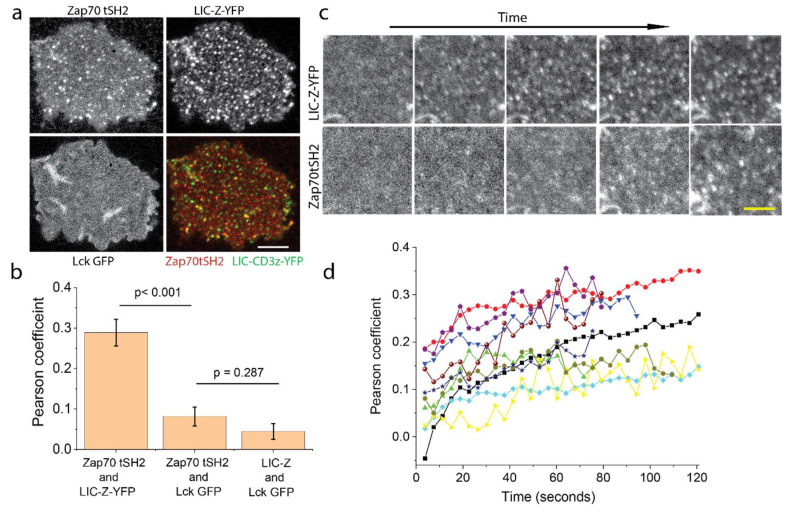
Zap70 tSH2 is recruited to LIC-Z-YFP clusters in reconstituted COS-7. COS-7 cells were reconstituted to co-express LIC-Z-YFP, mCherry-tagged Zap70 tSH2 and wild-type Lck GFP. (**a**,**b**) Representative images of Zap70 tSH2, LIC-Z-YFP and Lck GFP colocalization in COS-7 cells immediately fixed after 2 min irradiation with blue light (**a**) and corresponding Pearson coefficient analysis (**b**); Scale bar = 10 µm. In (**b**), the third column is Pearson coefficient values between LIC-Z and Lck GFP channel in COS-7 cells expressing only these two constructs. Data are mean and standard error of *n* = 20 cells. * *p* < 0.05, (one-way ANOVA with Fisher LSD post hoc test). (**c**,**d**). Representative images of the dynamic recruitment of Zap70 tSH2 to newly formed LIC-Z-YFP clusters in live COS-7 cells (**c**) and frame-by-frame Pearson coefficient values (**d**) between Zap70 tSH2 and LIC-Z-YFP channel; Scale bar = 3 µm. Each trace is one cell.

**Figure 7 ijms-21-03498-f007:**
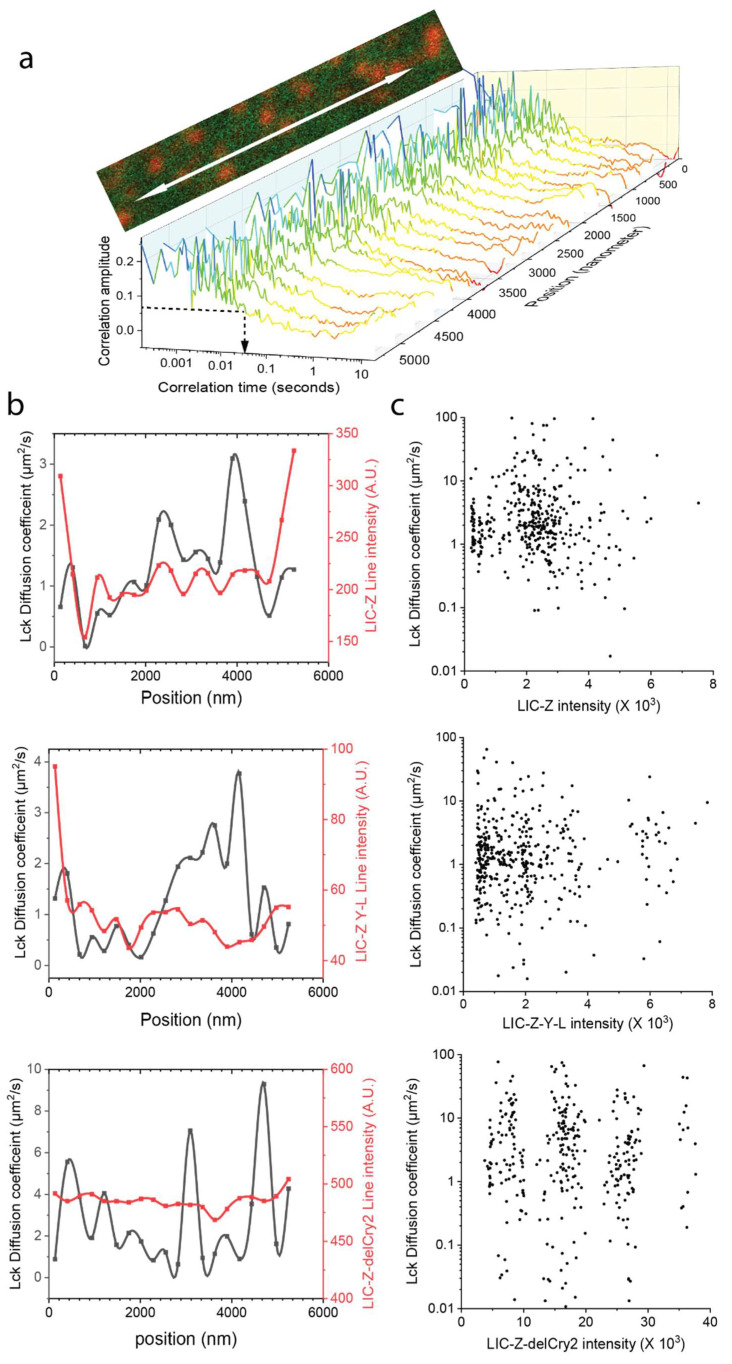
Diffusion analysis of Lck in the absence or presence of LIC-Z clusters by line-scanning fluorescence spectral correlation spectroscopy (FSCS) in COS-7 cells. ζ-chain signaling was reconstituted in COS-7 cells by co-expressing LIC-Z and wild-type Lck GFP. (**a**) Line-scanning FSCS was performed across the basolateral plasma membrane for 5.4 µm by 488 and 594 nm excitation as a bidirectionally scan at 1.3 kHz. A region of interest underneath the cell nucleus that contains LIC-Z clusters (red) and Lck GFP (green) as illustrated on the cropped image was chosen to perform the Line-scanning FSCS. The scanned line of the Lck channel was segmented into 20 equal parts for which the autocorrelation curves were plotted. Color indicates correlation amplitude, from which the correlation lag time of half the amplitude of G(0) was used to estimate average Lck diffusion time as indicated by black dotted arrow in the plot. (**b**) Representative line intensity profile (red line) of LIC-Z (top), LIC-Z-Y-L (middle) or LIC-Z-delCry2 (bottom) and estimated diffusion coefficients of co-expressed Lck GFP (black line) along the scanning line. *n* = 20 cells. (**c**) Scatter plot of 400 intensity values of LIC-Z (top), LIC-Z-Y-L (middle) or LIC-Z-delCry2 (bottom) versus estimated Lck diffusion coefficients of the same positions along the line.

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
