# Peer review of "Clustering of the ζ-Chain Can Initiate T Cell Receptor Signaling"

_ijms, 2020, doi:10.3390/ijms21103498_

Round 1
Reviewer 1 Report
As this paper is essentially identical to the one reviewed at eLife, I am not sure I can add much more than what was suggested previously to the authors. While the minor text changes based on the feedback and the dropping of explicitly stating that LIC-Z is monomeric are welcome, there remains the issue that no evidence is provided of the oligomeric status of LIC-Z prior to illumination. This would have been interesting; if LIC-Z is actually a small aggregate (<10) in the 'dark' but doesn't signal then this would argue against local (nano-) clustering being the main driver of receptor activation and would be a novel result.
The discussion states that their work agrees "with previous studies that found that only oligomeric forms of the ligands can trigger TCR activation [5,6,51]." This is rather disingenuous. Whilst it is true that soluble TCR ligands must be multivalent, when bound to a surface monomeric ligands are very efficient at triggering T cells. References to the significant number of papers that demonstrate this are not provided. Indeed, Jay Groves' lab have a nice paper in Biophys J this very week that directly shows a monomeric ligand is sufficient for triggering.
As no further experiments have been performed since the previous review, I restate that it would have been nice to know whether downstream signaling events could be induced by light (IL-2 production, CD69). It would also have been interesting to know whether LIC-Z clustering was reversible and so how efficiently signaling was diminished on return to the 'dark'. My inference is that the clustering was therefore irreversible but this is not discussed in the text.
Reviewer 2 Report
The authors used a in vitro system to investigate the role of CD3-zeta chain in TCR clustering and signaling. The biochemistry data had support the authors' hypothesis. Fundamentally, the authors still need to address more on the view of physiological process.
1 Is this zeta-chain clustering essential for TCR signaling initiation or it's more like a regulatory process?
2 What about the downstream gene expression upon the light induced zeta-chain clustering comparing to traditional TCR activation?
minor revision:
1 It's better to have a statistic analysis for figure 1C.
2 The knock out efficiency need to be shown for figure 2d.
Round 2
Reviewer 1 Report
Authors have done enough to now warrant acceptance